# User Performance in Virtual Reality Environments: The Capability of Immersive Virtual Reality Systems in Enhancing User Spatial Awareness and Producing Consistent Design Results

Sahand Azarby * and Arthur Rice

College of Design, North Carolina State University, Raleigh, NC 27607, USA
* Correspondence: sazarby@ncsu.edu

**Abstract:** Spatial decision-making in different virtual environments may vary based on the levels of spatial awareness experienced within Virtual Reality (VR) systems. Features and characteristics of VR systems may act as facilitators or distractors of spatial awareness, which can result in extreme variations in user spatial decisions. This research explored the capability of an Immersive Virtual Reality Interactive Environment (IVRIE) and a desktop-based VR (DT system) in reducing extreme variations in spatial decisions and inconsistent design results. Users' spatial decisions, performance, and design results in both systems were studied regarding the impact of these two systems' features on users, including the sense of immersion, types of interaction, and usage of eye-level view in spatial designs. The design results produced in both systems were compared considering the types of enclosure, surface texture, scale, and spatial function. Descriptive and inferential statistical comparisons and testing using quantitative and qualitative data were applied to determine participants' performance regarding the production of spatial outliers in each system. The results showed that IVRIE was more effective than the DT system fostering a consistency of space sizes and reducing outliers.

**Keywords:** Virtual Reality; immersive virtual reality systems; spatial design; spatial awareness; spatial decision-making; architectural design; consistent design outcomes; spatial outlier



## 1. Introduction

This study explored and compared user spatial awareness and spatial decision-making in two Virtual Reality (VR) systems, an Immersive Virtual Reality Interactive Environment (IVRIE) and a desktop-based VR (DT system). The goal was to determine the frequency of spatially abnormal design outcomes produced in each system. This paper is comprised of four sections: introduction, methodology, results, and discussion. The introduction includes the purpose statement, research questions, and research background, and a summary of a literature review focusing on previous studies on VR systems. The second section describes the chosen methodology and research design, including experiment design and applied methods for data collection. The results section presents the analyses and findings of this study. The discussion section presents the summary of the study's overall findings, limitations, and future implications.

### 1.1. Purpose Statement and Research Questions

The objectives of this study were to determine and compare user spatial decision-making in IVRIE and DT systems concerning spatially abnormal design outcomes in each system. The assumption was that the average number of users' spatial decisions with out-of-range design results (spatial outliers) would differ between IVRIE and DT systems. Additionally, the sequence of systems usage might decrease spatial outliers in the second system. The research questions were as follows:

- Does design in an IVRIE system, which creates the feeling of full immersion, allows for direct interaction with design objects, and provides an eye-level view, enable users to make more rational spatial design decisions than a conventional DT system?
- If users produce fewer space size outliers in IVRIE, is there any connection between their awareness of IVRIE's features and their spatial performance?
- Considering the production of spatial outliers in both IVRIE and DT systems, does systems usage sequence have any impact on the reduction of spatial outliers and production of out-of-range space sizes in the second system?

The hypotheses for this research study were as follows. First, the specific features of IVRIE, including full immersion, direct interaction, and access to eye-level viewpoints, facilitate users intuitive spatial decisions and the probability of decisions resulting in spatial outliers decreases. Second, using a combination of IVRIE or DT system will reduce spatial outliers. Third, the users who find IVRIE's features helpful for spatial design make less decisions resulting in spatial outliers.

*1.2. Research Background*

Virtual Reality (VR) is a computer generated three-dimensional (3D) environment in which users can have active experiences through the combination of the sense immersion and interaction [1,2]. VR provides accurate immersive visualization and creates a 3D interpretation environment in which users' mental representations created by their spatial perception become more coherent and enhance their understanding of spatial characteristics of complex models [3–5]. When using VR in design, 3D models become interactive models, which can be more visually complex and explorable by users walking within and around them and having a cognitive experience in an eye-level view [6,7]. VR enables users to be present within a design concept by allowing them to move through the design and test design ideas through intuitive interactions such as physical gestures [8–10]. Indeed, VR is a technology that, by adding dimensions to immersion and interactivity to 3D models, allows exploration of virtual environments on the human scale, which results in users' enhanced spatial cognition [11–13].

Virtual environments in the architectural profession and education are being applied for various purposes, including communication for the specifications of a design, finding suitable design solutions, providing interactive design experiments and enhancing design concepts perception and learning [7,14,15]. VR has the potential to represent designs with a higher perceptual accuracy that can impact visual understanding and design thinking positively and support design alterations when needed [16–18]. Sense of immersion is a critical factor in leading the users to experience an emotional and rational perception of 3D virtual environments [19,20].

Perception of the spatial nature of virtual spaces through a sense of immersion and scalability has made the immersive Virtual Reality (IVR) an important and emerging design tool. Increased awareness of the 3D characters and factors in design facilitates spatial cognition and can lead users to more rational design decisions and solutions [16,21]. Spatial awareness in VR systems enables users to overcome cognitive limitations and results in better estimations for spatial relationships in architectural design [19,22–25]. Research has also shown that among various digital tools/environments in architectural design, those that enhance spatial cognition of users positively impact the capacity of engagement between user and design, which results in higher awareness of spatial impacts of design decisions [21,26,27].

Spatial awareness in virtual environments is linked to the level of spatial presence that a user experiences based on the combination of IVR systems' features, including immersion and interaction [17,28,29]. The variation of sense of immersion and the types of interaction within IVR systems result in different perception levels of the spatial factors in design and awareness of spatial relationships [30–33]. Attention and involvement are two other factors that can impact the level of spatial presence in virtual environments. In most studies, attention is related to involvement and is defined as "the degree of significance or meaning

that the individual attaches to the stimuli, activities, or events" [34] and "whenever users pay more attention to the virtual environment stimuli, they become more involved in the virtual environment experience, and their sense of presence increases" [6]. Additionally, involvement as one of the spatial presence imperatives in virtual environments is linked to the degree and immediacy of control that a user has over the task environment or in interaction with the virtual environment and concluded that "having more control results in greater the experience of presence" [35].

IVR systems have been categorized into fully immersive and semi-immersive environments. The sense of immersion in fully immersive environments is higher than the semi-immersive ones since the sense of indirect interaction with design objects is replaced by direct interaction within them [36–40]. Additionally, it is mentioned that interaction is more natural in fully immersive virtual environments, which can then increase the intuitive sense of immersion and thus spatial perception and awareness [17,41–43].

In most studies, a comparison between fully immersive and semi-immersive environments shows an increasing sense of the presence for users in fully immersive. However, the usage of semi-immersive VR could have positive impacts on developing design ideas [20,25,38,44]. Some research concluded that the spatial awareness of users in semi-immersive VR systems is lower than in fully immersive ones since "they only partially enclose the viewer in any number of sensory inputs such as visual, physical and even audible" [25].

In the last decade, research focusing on the usability and capability of VR systems in architectural design education and practice has increased. In most studies, the scale and methods for comparing the efficacy of existing VR systems rely on qualitative data, mostly collected from users' self-evaluations and reports regarding their experience using the systems. Research applying quantitative or mixed methods is rare among recent studies. When VR systems are being studied to be identified as a more functional system based on the integration of user characteristics, systems features, and procedures of systems utilizations, opinion-based data cannot be used individually as a reliable source of data. This body of research with the goals of determining the efficacy and capability of VR systems in transferring spatial data to users and enhancing user understandings of spatial design imperatives needs to include quantitative data extracted from real design results.

Despite critical differences in the features of VR systems and how users apply them for design, fully immersive interactive and desktop-based VR systems are being used for spatial design purposes in architecture and related design categories. The findings in previous research showed that the spatial decisions of users in two different IVR systems, Immersive Virtual Reality Interactive Environment (IVRIE) and a desktop-based VR (DT system), regarding the differences in their features, characteristics of virtual spaces, and systems' usage sequence result in significant size variations in designed spaces between the two systems [31]. Additionally, in another study conducted by the authors, the findings declared that the integration of some users' backgrounds and spatial characteristics of virtual spaces act as perception filters. These filters impact users' spatial decision-making and result in the production of spaces with significant size variations utilizing these two IVR systems. Based on findings in previous research, another question was raised regarding the capability of these two systems in increasing consistency in spatial decision of users and therefore decreasing in the production of spatial outliers.

## 2. Methodology and Research Design

The methodological framework of this study is based on sequential mixed methods research comprising two phases of research design. The authors applied the current methodology in a previous study to identify the differences in users' spatial perception between the two VR systems and the impacts of systems utilization on spatial design decisions, and the production of design results with significant size variations [31]. The same methodology is applied in the current study with new research objectives, data types, and statistical analyses. This study's sequential mixed methodology comprises two phases

for collecting quantitative and qualitative data and analyzing each. Quantitative data was collected in the first phase to identify data dispersions. In the second phase, this quantitative data was used in qualitative data analyses by applying descriptive and inferential statistic testing. The collection of quantitative data relies on measuring the size of designed spaces created by the sample population in both IVRIE and DT systems. The qualitative data was extracted from the Spatial Perception Questionnaire (SPQ). This questionnaire gathers participant evaluations of helpfulness levels in spatial design of each system. The proposed method for collecting quantitative data is based on the authors' research and findings regarding producing accurate data suitable for statistical dispersions analytics. The chosen method for qualitative data collection relies on the recommended data gathering methods in within-subjects design studies, in which questionnaires are being utilized to collect required qualitative data for addressing research questions [45,46]. The collected qualitative data through SPQs were coded into numeric scales utilization in statistical comparisons concerning the quantitative data extracted from the participants' design results in IVRIE and DT systems.

The experiments for each participant had three steps. Each participant started work on the virtual model scenarios in one of the systems (i.e., DT system) and then moved to the second system (i.e., IVRIE) to work on the same models. After finishing the design of the last model in the second system, participants completed the SPQ.

Each participant completed the experiment by working on two sets of models in both systems. Each set (scenario) consisted of four spaces; two same-size enclosed spaces and two same-size corridor spaces. The difference between the two sets was that in one set, all the interior and exterior surfaces of all spaces had plain color with no texture, and in the other, all the space surfaces had a patterned texture. The participants were given four different spatial/experiential guidelines for redesigning the spaces. Participants used these four guidelines for two sets of models in the DT system and two sets of models in IVRIE. Each spatial/experiential guideline included the spatial function of a space, number of users to be accommodated, and a primary and secondary spatial feeling to be achieved.

The experiments were designed and conducted in four phases. In phase 1, systems setup, software selections, and testing and development of virtual models were completed. The SketchUp® software (Trimble Inc., Sunnyvale, CA, USA) was selected as a monitor–based VR environment (DT system). It was felt that this software has all the required features of a semi-immersive virtual environment for use in the study. VR Sketch® program, which works as a plug-in for various architectural software, including SketchUp®, was selected as a fully immersive environment for use in the Immersive Virtual Reality Interactive Environment (IVRIE). Participants interacted with the fully immersive environment through the usage of a head-mounted device.

The DT system as a monitor-based VR system was a high-performance computer system equipped with a 32" full-HD monitor for the presentation of the VR environment and a mouse device as an interaction interface between the VR environment and the user. For IVRIE, the VR Sketch® program as a fully immersive environment was provided through an Oculus Quest 2 device set, including a VR goggle headset and two touch controllers for interacting with virtual objects in the environment. Each participant used the DT system by sitting at a desk and worked in IVRIE by putting on the headset and being able to walk and move in a safe zone designated for the IVRIE section of the experiments. The virtual models utilized in the experiments were developed in SketchUp® software, and two sets of models, each consisting of two same-size corridor spaces and two same-size enclosed spaces, were built. The sequence of spaces in both sets was corridor 1, enclosed 1, corridor 2, and enclosed 2, and all spaces had plain texture in the first set and patterned texture in the second set. The starting width for all corridors was 10 feet, and the starting dimension of all enclosed spaces was 10 feet by 10 feet (100 sq. ft.). In the DT system, participants manipulated and redesigned the models directly within SketchUp® software, and in IVRIE, they did the same for the models exported into the VR Sketch® extension.

The sample population was divided into two groups. One group started working on both plain and patterned sets of models in the DT system and then moved to IVRIE to work on the two other sets of models. The other group completed the experiment in the opposite order. Systems usage sequence as a variable that was tested to determine if usage sequence would have an impact on increment or decrement of designing out–of–range space sizes resulting from extreme spatial decision-making. Each participant started redesigning the spaces with the plain scenario and then moved to the patterned scenario, either working in IVRIE or DT system. All participants spatial decisions attempted to follow the spatial/experiential guidelines. Figure 1 presents the experiment setup and systems usage sequence.

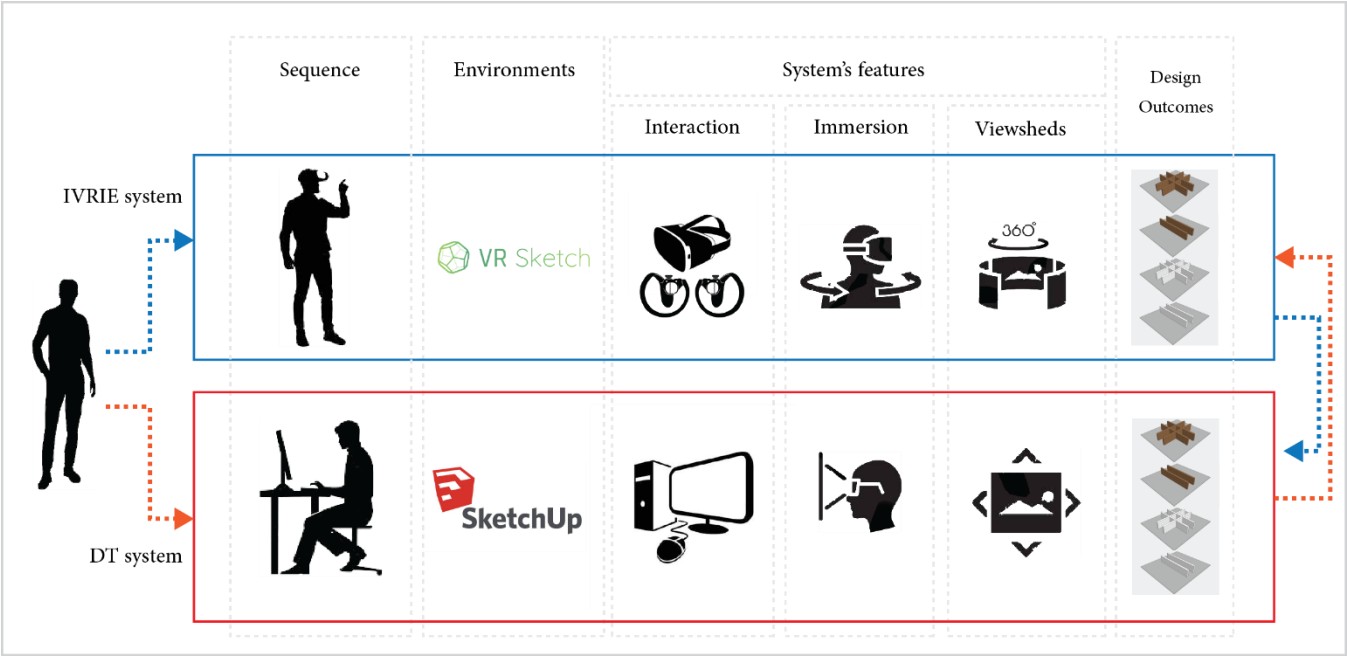

**Figure 1.** Experiments setup scheme and systems usage sequence.

In this study, the reason for developing two sets of models to be used in both systems and presenting the virtual spaces with two different types of surfaces (plain and patterned) to the participants was to determine the impact of surface texture variations on spatial decision-making. Figure 2 presents developed models with their initial sizes.

In the second phase of experiment design, the modification and finalizing of experiential/spatial guidelines was completed, along with the refining of SPQ questions. In addition, participant recruitment was completed at that time. There were four different spatial/experiential guidelines developed. Two guidelines were for redesigning the corridor spaces and two for enclosed spaces. Each guideline was designed to give the spatial/experiential information for redesigning a space and included the number of people to be accommodated in the space, designated function of the space, and spatial feeling of the space (See Appendix A). Each participant used guidelines for redesigning the four spaces in the first set of models (plain scenario) and then used the same set for redesigning the four spaces in the second set of models (patterned scenario) while working in the DT system. The same process of applying guidelines to redesign the spaces was repeated for all of the same models in IVRIE.

The SPQ consisted of five questions. In the first three questions, participants were asked to evaluate the helpfulness levels of IVRIE features in their spatial decision-making including direct interaction with design objects within the virtual environment, using eye-level view for observing the surroundings, and feeling immersed within the virtual environment. The fourth question was focused on systems usage sequence. The goal was

to get participants' evaluations regarding the helpfulness of getting familiar with spaces and guidelines in the first system and using this familiarity with spatial decision-making in the second system. Participants designated their evaluations regarding system features and usage on a Likert-scale of helpfulness levels from "Very helpful" to "Not at all helpful". In The last question on SPQ, participants evaluated the overall functionality of one or both systems, allowing them to effectively create the intended spatial sense based on spatial/experiential guidelines (See Appendix B).

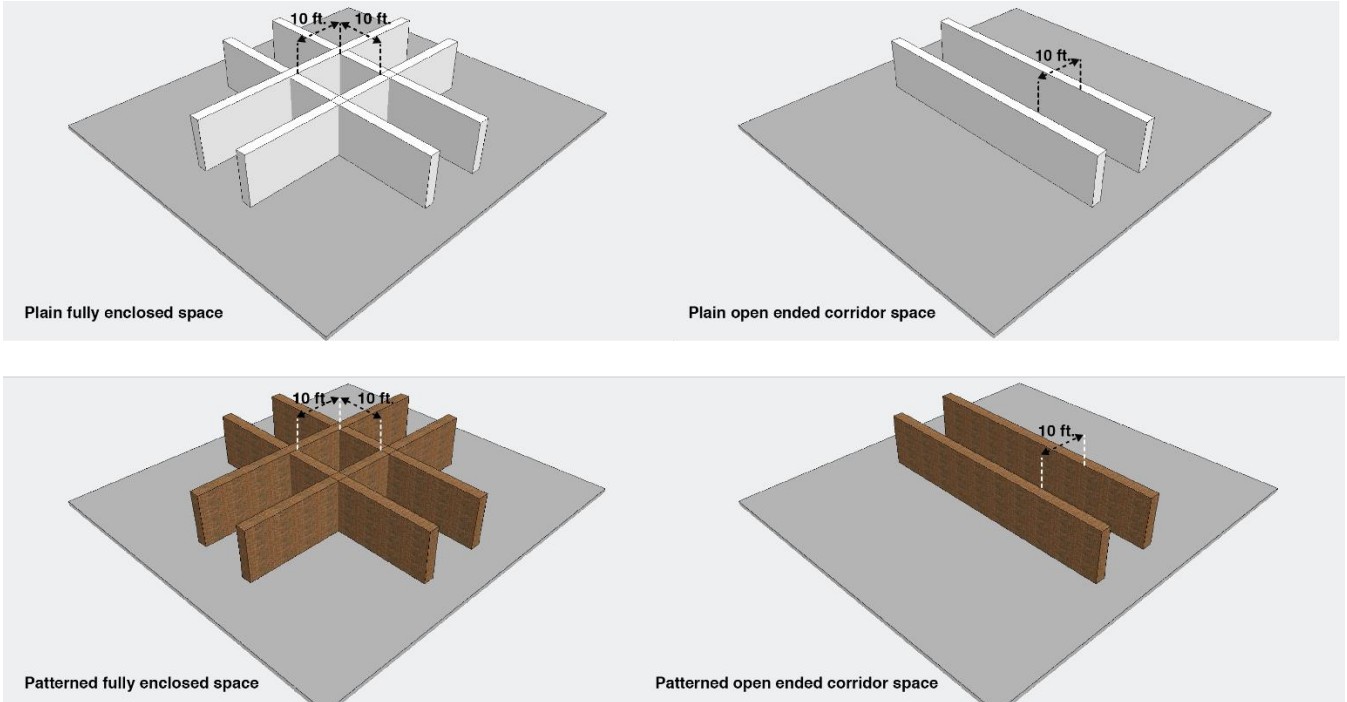

**Figure 2.** Developed models for experiments.

For this study, 65 participants were recruited, including design students in architecture, landscape architecture, industrial design and engineering students in civil engineering and computer science. The age range of participants was between 20 to 25 years old, and the sample population consisted of 31 male and 34 female participants. All participants were familiar with the SketchUp® software, and all had the experience of using head-mounted devices at least once.

In phase 3, data was collected one participant at a time. Data collection from each participant started with recording design results in both systems and was followed by gathering the SPQ answers to the questions. Regardless of participants' systems usage sequence, the design results of all participants, consisting of 8 redesigned spaces in the DT system and 8 redesigned spaces in IVRIE, were collected and saved as skp files. From the measurements of spaces produced by each participant, utilizing both systems, 16 numeric values were extracted consisting of the sizes of 4 corridor spaces and 4 enclosed spaces designed in the DT system and 4 corridor spaces plus 4 enclosed spaces designed in IVRIE. Data extracted from each participant's answers on the SPQ was coded into numeric scales for sample population statistical comparisons. Data analyses and descriptive and inferential statistical testing were completed in phase four. Figure 3 presents the sequence of experiment plan phases.

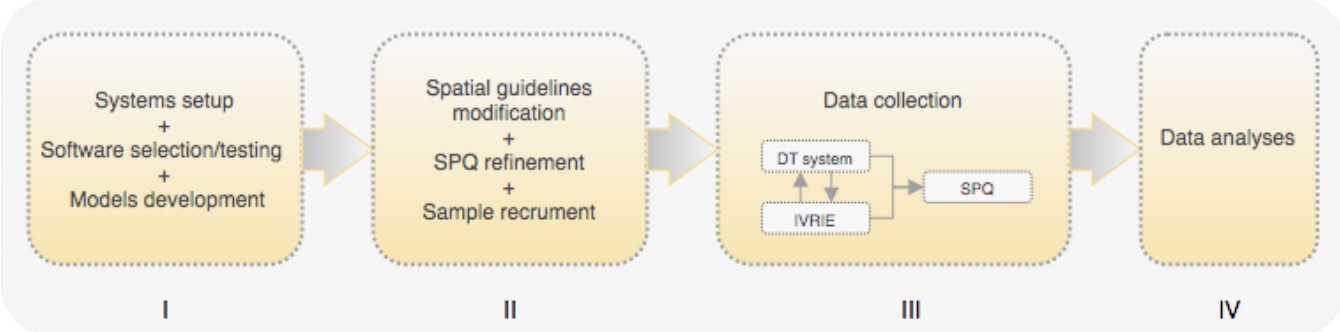

**Figure 3.** Flow chart of the experiment design.

### 3. Results

Data analyses and descriptive statistics for this study are structured using the utilization and calculation of interquartile range (IQR), which is a measure of statistical dispersion and spread of quantitative data collected from the sample population. IQR is calculated for the size of all types of spaces designed by participants in IVRIE and DT systems. The calculation of IQRs for corridors spaces and enclosed spaces designed by participants identifies outliers, which are spaces produced with out-of-range sizes. The outliers are data points that differ significantly from other observations and have values that lie at an abnormal distance from all the others. The abnormal distance of a data point from others results from lying of that data point below the lower limit or above the upper limit of the data range. For this study, when the width of a corridor or the size (inner area) of an enclosed space designed by a participant is significantly wider/narrower or larger/smaller compared to the width and size of the corridors and enclosed spaces in the same category (same enclosure type, texture, and spatial function) designed by sample population, it has been detected as an outlier.

The results and analyses of this study are categorized into two branches. The first branch comprises the statistical analyses of total out-of-range design results percentages produced by participants in each system. The out-of-range design results analyses compare the percentage differences of produced outliers for each space between the two systems considering the space types (open-ended corridors and fully enclosed spaces), presence or absence of surface texture, the designated function of spaces, systems usage sequence and user characteristics including gender and educational major of participants. The produced spatial outliers utilization of both systems by participants as dependent variables were analyzed through the application of Cramer's V test to identify the strength of association with other variables, including spaces' characteristic, users' characteristics and systems usage sequence. Cramer's V is a normalized version of the Chi-square test and measures the strength of association between two nominal variables.

The second branch of analysis compares the percentage of out-of-range design results in each system based on the sample populations' perception of the systems' ability to facilitate participants in making more accurate spatial decisions. The correlations between the produced outliers in both systems and participants' perceived helpfulness levels of IVRIE's features utilization and systems efficiency in spatial decision-making were tested and analyzed using the Spearman correlation test.

#### 3.1. Percentage of Out of Range Design Results within Each System

The calculation of the percentage of produced outliers in both systems determined that of 1040 designed spaces (65 participants, each with 16 designed spaces, divided equally in both systems), 3.85% of spaces were detected as outliers with 2.31% resulting in DT system, and 1.54% resulting in IVRIE. This comparison shows that the spatial decisions of participants in the DT system, regardless of the type of spaces, their spatial characteristics, or the sequence of systems usage, resulted in producing a higher percentage of space size

outlier compared to IVRIE. Figure 4 presents the total percentage of produced space size outliers in each system.

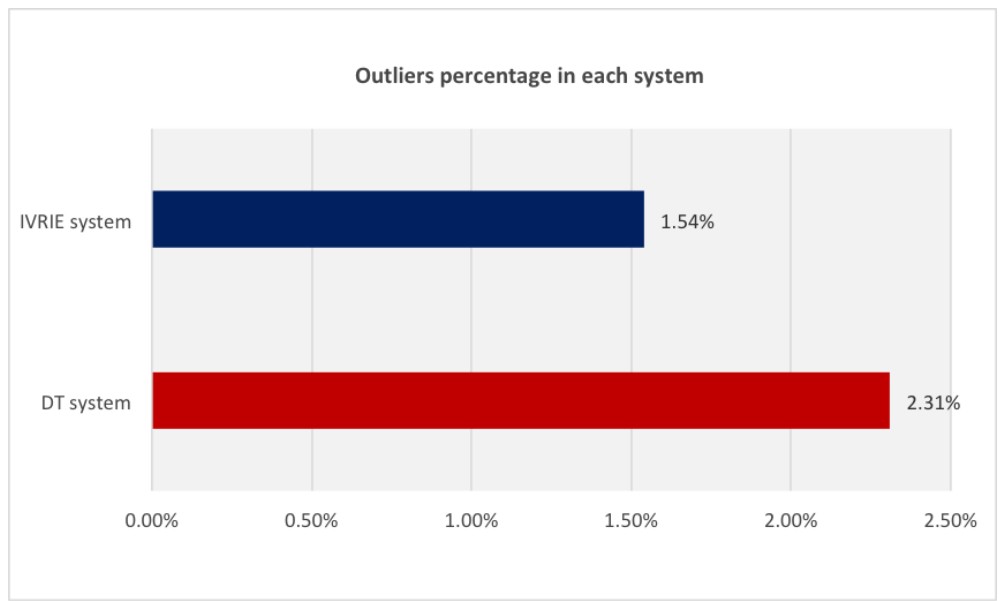

**Figure 4.** Total percentage of produced space size outliers in each system.

The inspection of produced out-of-range space sizes in both systems based on the types of space found that the percentage of outliers in open-ended corridors in both systems is 32.5% and for fully enclosed spaces is 67.5%. The higher percentage of enclosed space outliers in both systems indicates that the increment of design objects in the structure of a space (two walls in a corridor compared to four walls in an enclosed space) makes spatial decisions for the logical size of the space more challenging. Although the total percentage of fully enclosed spaces as spatial outliers was higher than open-ended corridors utilizing both systems, there is not a strong association between the production of out-of-range sizes of both space types and the utilized systems. The Cramer's V of 0.02 declares a weak association between the space types and utilized systems in producing out-of-range space sizes. Figure 5 presents the total produced outliers in both systems based on the type of space.

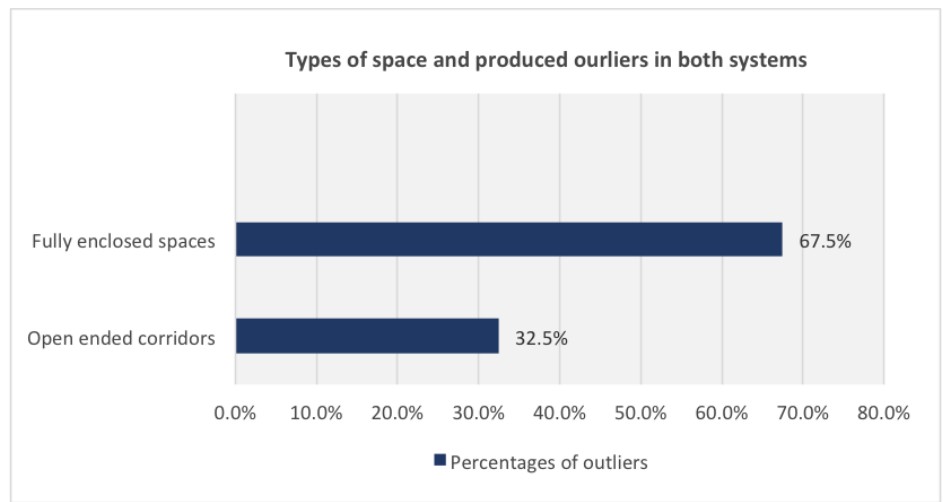

**Figure 5.** The percentage of total produced outliers for each space type in both systems.

The comparison of produced out-of-range space sizes utilizing both systems based on participants' characteristics, including gender and educational majors, declared that the percentage of outliers in each system, regardless of the spaces' characteristics and systems usage sequence varied. Based on gender categories, male participants (48% of the sample population) produced more out-of-range space sizes than female participants (52% of the sample population) using the DT system or IVRIE. The percentages of produced outliers by male participants in DT system and IVRIE were 5% and 10%, respectively, compared to female participants. Although the comparison between the outliers percentages declares that the overall performance of female participants was better in both systems, there was not a strong correlation between gender differences and systems variation in the production of out-of-range space sizes. Cramer's V of 0.08 declares a weak association between gender and utilized systems in producing out-of-range space sizes. Of the sample population, 35% of the participants' major was architecture, 32% were landscape architecture, and 32% had other majors, including industrial design, civil engineering, and computer science. The comparisons of produced outliers by each 'major' category utilization of both systems declared that participants in architecture major produced equal percentages of outliers either using DT system or IVRIE. The participants in the landscape architecture major produced 8% fewer outliers in DT system compared to IVRIE. The participants in the 'other majors' category produced the highest percentage of out-of-range space sizes in DT system (38%) compared to the architecture and landscape architecture major categories, and the difference in the percentages of their produced out-of-range spaces between the two systems was 18%. According to the correlation between the educational majors and utilized systems in the production of outliers in each system, Cramer's V of 0.39 declares a moderate association between major categories and utilized systems in producing out-of-range space sizes. Figure 6 presents the out-of-range space sizes in both systems concerning participants' gender and major categories.

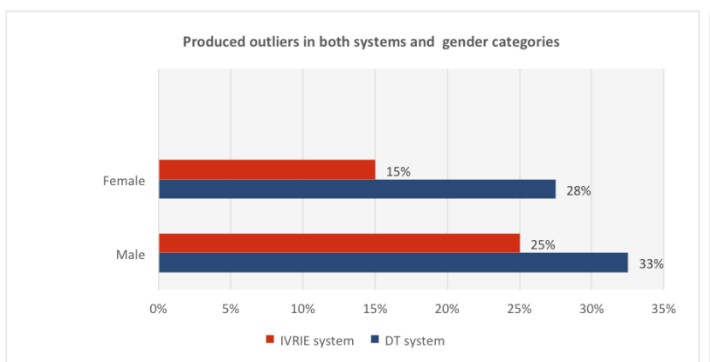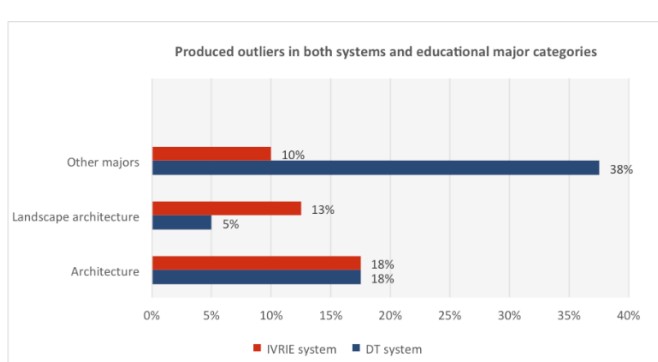

**Figure 6.** The percentage of produced outliers in each system concerning gender and educational major categories.

The comparison of open-ended corridor spaces regarding the number of people walking down the corridor found that the difference between the out-of-range corridors produced for one-person walking and three people walking was 1% between the systems. The percentage of out–of–range corridors for one-person walking was 38% in the DT system versus 40% in IVRIE, and the corridors for three people walking were 63% in the DT system and 60% in IVRIE.

Comparing the percentages of produced out-of-range fully enclosed spaces between the two systems showed that the percentage of outliers for enclosed spaces accommodating the gathering of two people was 50% in the DT system versus 18% in the IVRIE. On the contrary, the percentage of produced out-of-range enclosed spaces accommodating the gathering of 10 people was higher in IVRIE compared to DT system and was 82% versus 50%. Regarding the correlation between space types and utilized systems in the production of outliers, Cramer's V of 0.02 declares a weak association between the space characteristics,

including type and spatial function, and utilized systems in the production of out-of-range space sizes. Figure 7 presents the percentages of outliers for both space categories between the two systems.

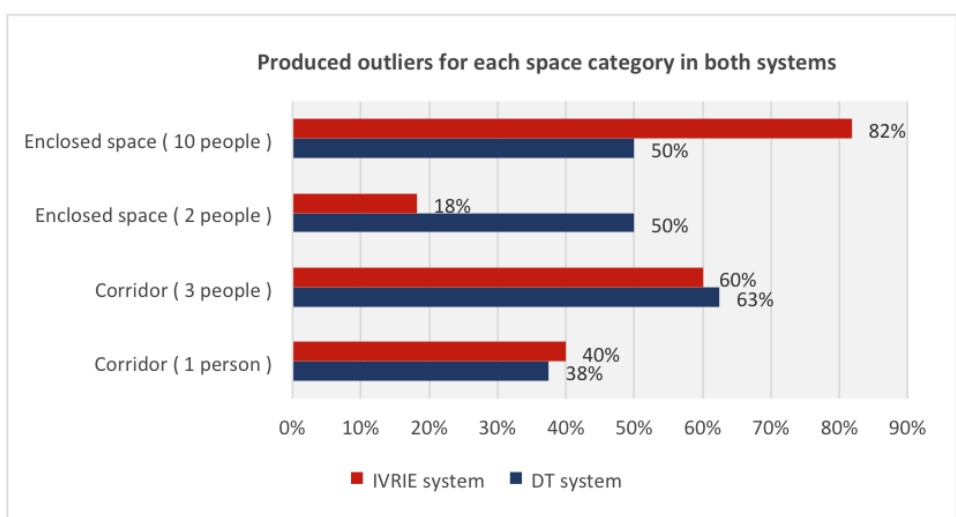

**Figure 7.** The percentage of produced outliers for each space based on the spatial function/scale between the systems.

The absence or presentation of texture had different impacts on the percentage of produced out-of-range sizes for both corridors and enclosed spaces between the two systems. The percentage of outliers for plain corridors for one person walking in the DT system was 37.5%, while participants in IVRIE produced no outliers for this type of space. On the other hand, the percentage of out-of-range patterned corridors for one-person walking in IVRIE was 40% and 0% in the DT system. The percentages of produced outliers for plain corridors for three people walking were close, and they were 50% and 60% in DT system and IVRIE, respectively. No outlier was produced for patterned corridors for walking three people in IVRIE, while 12.5% of designed patterned corridors for three people walking in DT system had out-of-range sizes.

The inspection of the number of produced out-of-range space sizes for fully enclosed spaces between the two systems showed that the percentage of outliers for enclosed spaces accommodating the gathering of two people, either in plain or patterned texture in IVRIE, was equal, with 9.1% outliers for these spaces. In the DT system, the percentage of plain enclosed spaces for gathering two people was 37.5%, three times higher than patterned enclosed spaces with 12.5% outliers. The percentages of produced out-of-range space sizes for plain, fully enclosed spaces accommodating the gathering of 10 people were close in both systems, and there were 31.3% and 27.3% in DT and IVRIE, respectively. The percentage of outliers for patterned enclosed spaces accommodating the gathering of 10 people in IVRIE was 54.5%, higher than the produced out-of-range spaces in DT with 18.8% outliers. The correlation testing between the presence/absence of texture regardless of the type of spaces (fully enclosed spaces or open-ended corridors) and utilized systems resulted in the Cramer's V of 0.31, which declares a moderate association between the presentation or absence of texture and utilized systems in producing out-of-range space sizes. Table 1 presents the percentage of outliers for all of the spaces in both IVRIE and DT systems. Figures 8 and 9 illustrate the outliers, upper/lower limits, and median size for each space category in both systems.

**Table 1.** Percentage of outliers for each space in both systems.

| Space | Texture | Capacity | Produced Outliers in Each System | |
|---|---|---|---|---|
| | | | DT | IVRIE |
| Fully Enclosed Space | Plain | 2 | 38% | 9% |
| | | 10 | 31% | 27% |
| | Patterned | 2 | 13% | 9% |
| | | 10 | 19% | 55% |
| Open Ended Corridor | Plain | 1 | 38% | 0% |
| | | 3 | 50% | 60% |
| | Patterned | 1 | 0% | 40% |
| | | 3 | 13% | 0% |

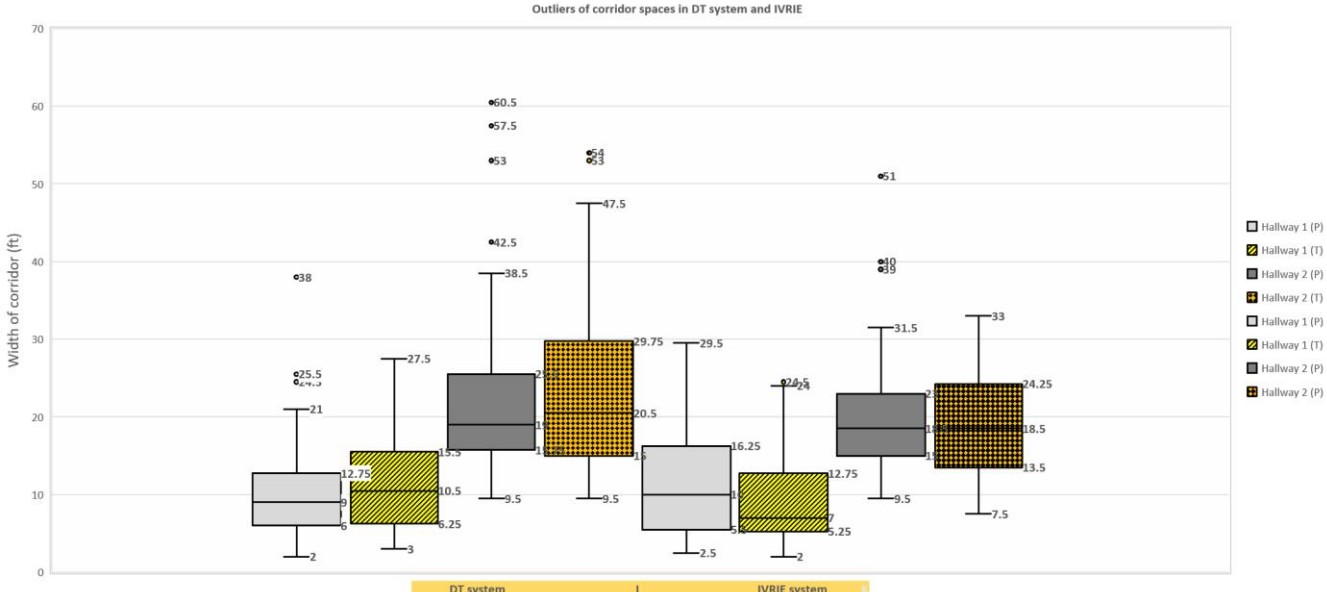

**Figure 8.** Outliers of plain and patterned corridors in both systems.

*3.2. Percentage of Space Size Outliers and Systems Efficiency Evaluation*

This branch of analysis identifies the percentage of outliers produced in each system concerning participants' self-evaluations of systems' features, usage sequence, and efficiency. The data for this analysis were collected from the participants' answers to the questions on SPQ and identified outliers from the produced design results by participants utilizing each system. Among five questions on SPQ, in three questions, participants were asked to evaluate the helpfulness level of IVRIE's features, including direct interaction, sense of full immersion, and having eye-level view in their decision makings for the scale of spaces. The fourth question pointed to the helpfulness level of systems usage sequence in better estimating spaces' scale. In the fifth question, all the participants evaluated the systems' overall capability to create the intended spatial sense of spaces more accurately. Based on the chosen answer options of participants, the sample was divided into subgroups. Then, the percentage of outliers produced by each group in each system was calculated to identify the compatibility of users' self-evaluation of systems efficiency and their real design results utilizing each system. The Spearman correlation test was the utilized statistical test for establishing the relationship between the numbers of produced outliers in each system by the participants, categorized by their perceived helpfulness levels of IVRIE's features and systems' usages. The Spearman correlation test measures the strength and direction of a monotonic association between two ranked variables in which the correlation coefficient declares the strength of the relationship between variables, and the *p*-value

identifies if the result of an experiment is statistically significant. By setting α = 0.05, the achieved statistically significant Spearman rank-order correlation means there is less than a 5% chance that the strength of the relationship between the produced outliers in each system by a portion of the sample population happened by chance if the null hypothesis was true. A null hypothesis for Spearman's correlation test was 'H0: There is no [monotonic] association between the produced spatial outliers in DT system and IVRIE in the population of evaluating groups.' The results of the analyses are as follows:

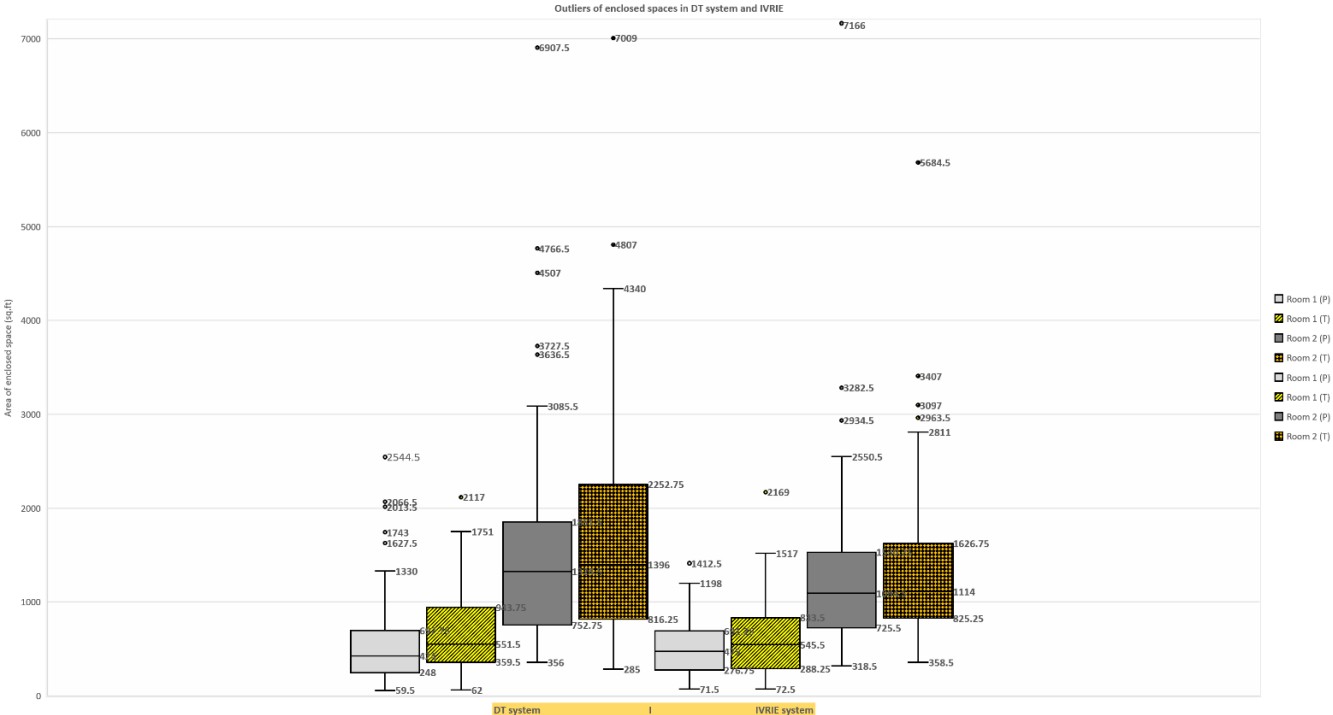

**Figure 9.** Outliers of plain and patterned enclosed spaces in both systems.

3.2.1. Helpfulness Level of Direct Interaction with Virtual Objects in IVRIE

In the first spatial question, the participants had to evaluate the helpfulness level of having direct interaction with design objects in IVRIE utilizing the VR headset handles (controllers). Of the sample population, 80% of participants chose "Very helpful," 17% "Somewhat helpful," 3% "Slightly helpful," and 0% "Not at all helpful." The percentages of produced outliers for each group in both IVRIE and DT systems showed that the group 1 (Very helpful level) had 28% more outliers in the DT system than in IVRIE. Group 2 (Somewhat helpful level) had 34% more outliers in the DT system than IVRIE. Group 3 (Slightly helpful) produced all their outliers in IVRIE and group 4 (Not at all helpful) had no outliers in their produced design results either using IVRIE or DT system. The calculation of Spearman's correlation coefficient and subsequent significance testing for the produced spatial outliers by the group with a perceived helpfulness level of "very helpful" for having direct interaction with design objects in IVRIE indicates a weak positive correlation coefficient ($r_s = 0.35$). The calculated *p*-value (0.009, significant) indicates that there is less than a 5% chance that the strength of this relation happened by chance if the null hypothesis was true. The correlation test for the group with the perceived helpfulness level of direct interaction in IVRIE "Somewhat helpful" indicates a weak positive correlation coefficient ($r_s = 0.37$) between the produced outliers in both systems. The calculated *p*-value (0.2, non-significant) indicates that there is more than a 5% chance that the strength of this relation happened by chance if the null hypothesis was true. The correlation test for group 3 (Slightly helpful level) could not be run because of the inadequate number of members.

Figure 10 illustrates the population of each group and their produced outliers percentage in both systems.

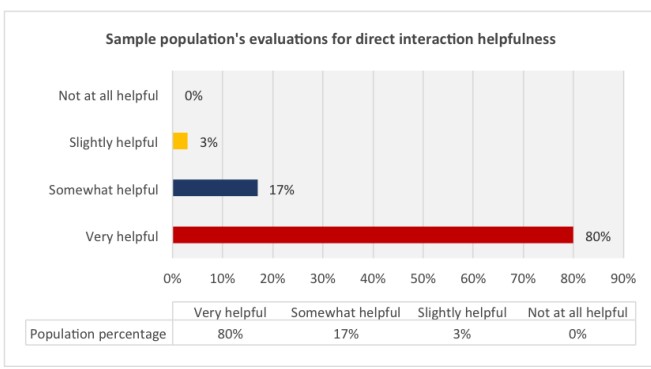 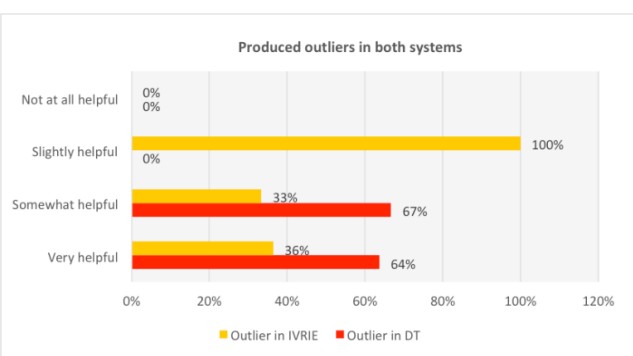

**Figure 10.** Perceived levels of helpfulness of direct interaction in IVRIE and produced outliers in both systems.

3.2.2. Helpfulness Level of Access to Eye-Level View in IVRIE

In the second spatial question, the participants had to evaluate the helpfulness level of having eye-level view in reaching the designated spatial feeling through designing the spaces utilizing the IVRIE. Of the sample population, 92% of participants chose "Very helpful," 6% "Somewhat helpful," 2% "Slightly helpful," and 0% "Not at all helpful." The percentages of outliers for each group showed that group 1 (Very helpful level) had 30% more outliers in IVRIE. Group 2 (Somewhat helpful level) produced all their outliers in the DT system than IVRIE. Group 3 (Slightly helpful level) did not produce any outliers either utilizing IVRIE or DT system.

The results indicate that group 1, who perceived the utilization of eye-level view in IVRIE "Very helpful" (92% of the sample population), produced a higher percentage of outliers in DT system. This could be interpreted that eye-level view utilization has an active role in reducing produced spatial outliers in IVRIE compared to DT system for most participants. The correlation test for this group indicates a weak positive correlation coefficient ($r_s$ = 0.33) between the produced outliers in both systems, and the calculated *p*-value (0.007, significant) indicates that there is less than a 5% chance that the strength of this relation happened by chance if the null hypothesis was true. The correlation test for group 2 (Somewhat helpful level) could not be run as the reason of lacking produced outliers in DT system. Figure 11 presents the population of each group and their produced outliers percentage in both systems.

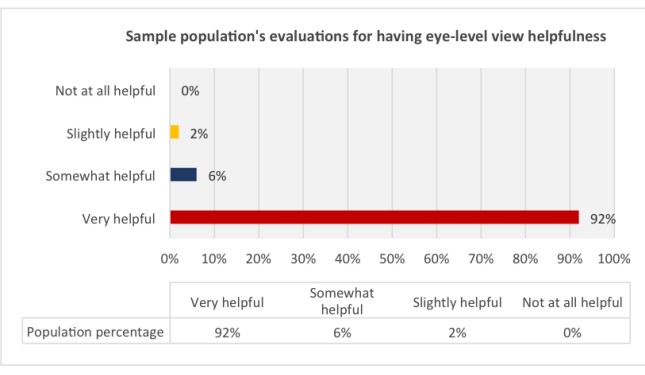 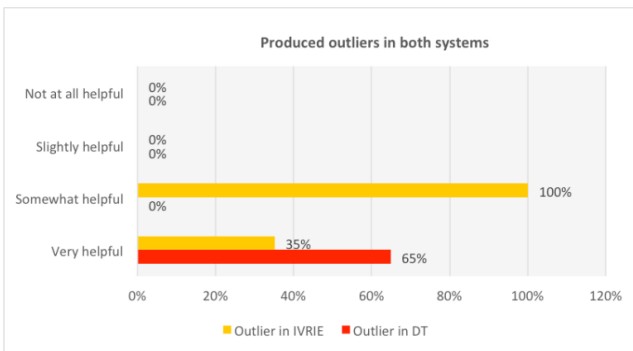

**Figure 11.** Perceived levels of helpfulness of eye-level view usage in IVRIE and produced outliers in both systems.

### 3.2.3. Helpfulness Level of Experiencing Full Immersion in IVRIE

In the third spatial question, the participants evaluated the helpfulness level of experiencing full immersion in understanding the feeling of the spaces in making spatial decisions utilizing IVRIE. Of the sample population, 80% of participants chose "Very helpful", 20% "Somewhat helpful", and 0% either "Slightly helpful" or "Not at all helpful". The percentages of outliers for each group showed that group 1 (Very helpful level) had 18% more outliers in the DT system than in IVRIE. Group 2 (Somewhat helpful level) had 34% more outliers in the DT system than IVRIE. The results show that participants with perceived high levels of helpfulness in experiencing full immersion in IVRIE (100% of the sample population) produced fewer out-of-range space sizes in this system compared to DT system. Indeed, the sense of full immersion within IVRIE had an impact on the reduction of produced spaces as outliers, and it is compatible with participants' overall awareness of the role of this feature in decision-making for the scale of spaces based on utilizing spatial/experiential guidelines. The correlation test for the group with a perceived helpfulness level of "very helpful" for experiencing full immersion in IVRIE indicates a weak positive correlation coefficient ($r_s$ = 0.37) between the produced outliers in both systems. The calculated *p*-value (0.006, significant) indicates that there is less than a 5% chance that the strength of this relation happened by chance if the null hypothesis was true. For the group with a "Somewhat helpful" evaluation level, there is a weak negative correlation coefficient ($r_s$ = −0.02) between the produced outliers in both systems. The calculated *p*-value (0.9, non-significant) indicates that there is more than a 5% chance that the strength of this relation happened by chance if the null hypothesis was true. Figure 12 illustrates the population of each group and the percentage of produced outliers in each system.

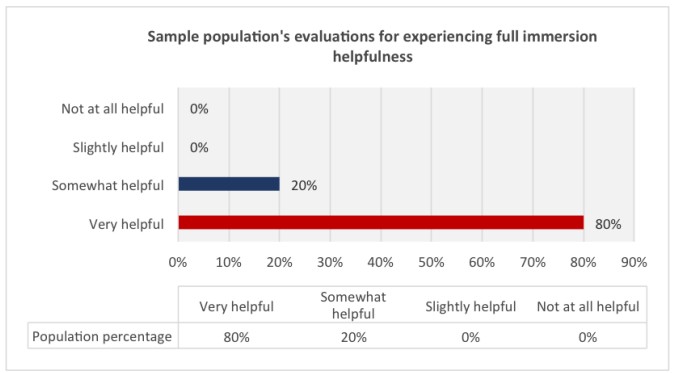 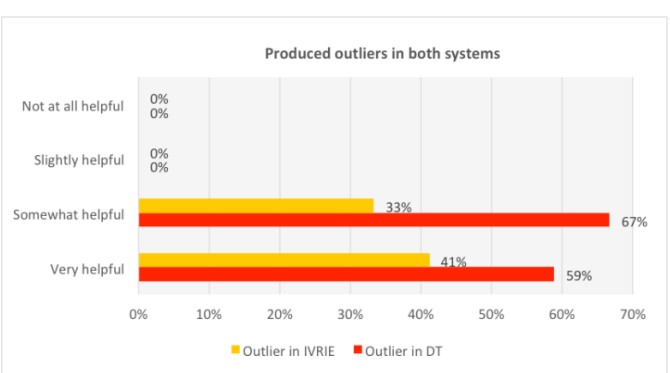

**Figure 12.** Perceived levels of helpfulness of experiencing full immersion in IVRIE and produced outliers in both systems.

### 3.2.4. Helpfulness Level of Systems Usage Sequence in Spatial Decision-Making

Based on the fourth spatial question, the participants evaluated the helpfulness level of systems usage sequence in becoming familiar with spaces in one system and using this spatial familiarity in the second system for spatial decision-making. The sample population was divided into two groups with roughly equal participants (51% and 49% of the sample population, respectively). The first group started the experiments with the DT system and then moved to IVRIE to complete the experiment. The second group completed the experiments in opposite direction; IVRIE first and DT the second. The inspection of produced out-of-range space sizes by each group in both systems found that group 2 with system usage sequence from IVRIE to DT system produced 20% fewer outliers compared to the first group. The total percentage of produced outliers in both systems by the group with systems usage sequence of DT to IVRIE was higher than the group with opposite systems usage sequence. However, there is no strong correlation between the production of spatial outliers in both systems and the variation in systems usage sequence. The Cramer's V of 0.14 declares a weak association between the variation of systems usage sequence in

the production of out-of-range space sizes utilizing both IVRIE and DT system. Figure 13 presents the produced outliers by each group in both systems.

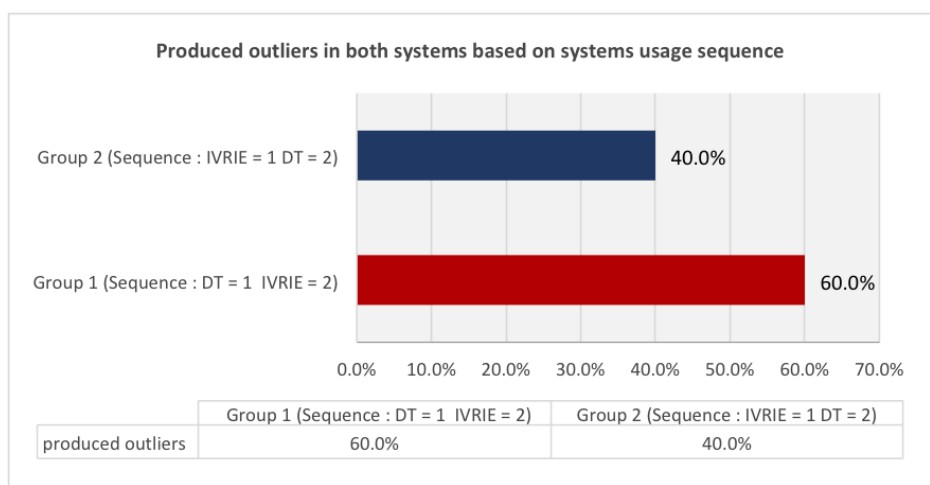

**Figure 13.** Produced outliers by each group with opposite system usage sequence in both systems.

The comparisons of the participants' evaluations for helpfulness level of systems usage sequence show that the second group with the system usage sequence from IVRIE to DT found this sequence more helpful in getting familiar with spatial characteristics of spaces in IVRIE and then using this familiarity for spatial decision-makings in DT compared to the first group with reversed systems usage sequence. 42% of participants in group 1 evaluated the helpfulness level of the system usage sequence "Very helpful", compared to 34% of participants in group 2. In both groups, the percentage of participants choosing the "Somewhat helpful" level was close, with 45% and 44% in groups 1 and 2, respectively. The percentage of participants in group 1 with an evaluation level of "Slightly helpful" was 10% less than in group 2, and it was 12% compared to 22% in group 2. Figure 14 presents the population percentage for each group's chosen helpfulness levels of systems usage sequence.

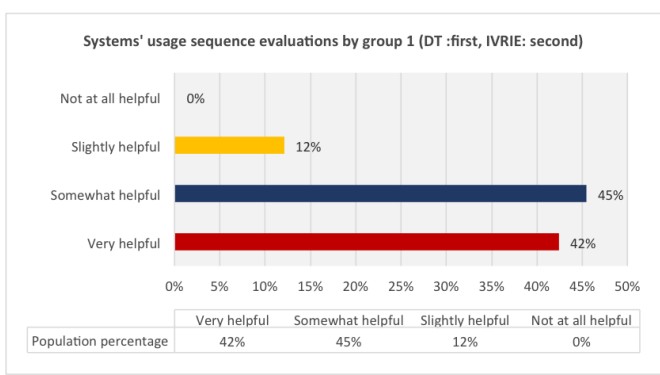
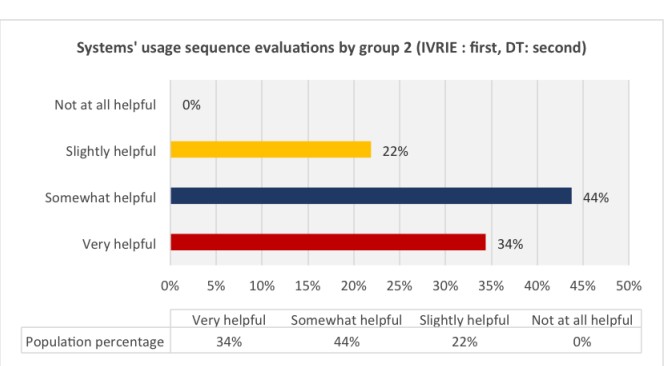

**Figure 14.** Perceived levels of helpfulness of systems usage sequence in each group.

The inspection of the produced outliers in both systems by each group shows that the participants in both groups who evaluated the systems' usage sequence helpfulness level "Very helpful" produced fewer outliers in IVRIE compared to DT. This result indicates that using IVRIE either as the first or second system still decreases the number of produced out-of-range space sizes. In addition, the percentage of outliers produced in DT system by the participants who used DT as the second system was 8% less than those with systems usage sequence of DT system to IVRIE. In both groups, the participants who evaluated the helpfulness level of systems usage sequence "Somewhat helpful" produced more outliers

in the second system based on their systems sequence utilization. In group 1 with system usage sequence from DT to IVRIE, these participants produced 72% more outliers in IVRIE compared to DT system. On the contrary, in group 2 with systems usage sequence from IVRIE to DT, the participants produced 61% more outliers in DT system.

The calculation of Spearman's correlation coefficient and subsequent significance testing for the produced spatial outliers by group 1 (system usage sequence: DT first, IVRIE second) with a perceived helpfulness level of "very helpful" for systems usage sequence in spatial decision-making indicates a moderate positive correlation coefficient ($r_s$ = 0.49). The calculated *p*-value (0.07, non-significant) indicates that there is more than a 5% chance that the strength of this relation happened by chance if the null hypothesis was true. The correlation test for the participants in this group with the perceived helpfulness level of "Somewhat helpful" indicates a moderate positive correlation coefficient ($r_s$ = 0.48) between the produced outliers in both systems. The calculated *p*-value (0.06, non-significant) indicates that there is more than a 5% chance that the strength of this relation happened by chance if the null hypothesis was true.

The calculation of Spearman's correlation coefficient and subsequent significance testing for the produced spatial outliers by group 2 (system usage sequence: IVRIE first, DT second) with a perceived helpfulness level of "very helpful" for systems usage sequence in spatial decision-making indicates a moderate positive correlation coefficient ($r_s$ = 0.57). The calculated *p*-value (0.06, non-significant) indicates that there is more than a 5% chance that the strength of this relation happened by chance if the null hypothesis was true. The correlation test for the participants in this group with the perceived helpfulness level of "Somewhat helpful" indicates a weak negative correlation coefficient ($r_s$ = −0.02) between the produced outliers in both systems. The calculated *p*-value (0.9, non-significant) indicates that there is more than a 5% chance that the strength of this relation happened by chance if the null hypothesis was true. Figure 15 presents the percentage of produced outliers in both systems for each group based on participants' evaluation levels.

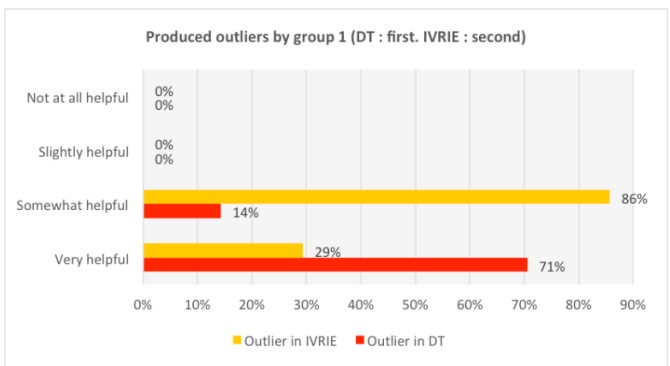 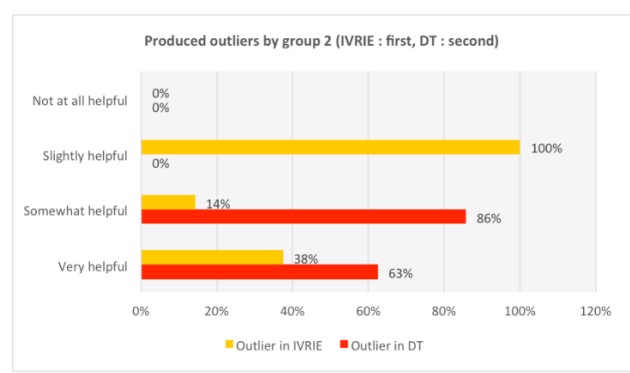

**Figure 15.** Perceived levels of helpfulness of systems usage sequence and produced outliers in both systems.

3.2.5. Systems' Effectiveness Levels for Accurate Spatial Decision-Makings

In the last spatial question, the participants were asked to evaluate the effectiveness of systems in allowing them to create the intended spatial sense of spaces with higher accuracy. Of the sample population, 3% chose the DT system, 80% selected IVRIE, and 17% indicated both systems. The percentages of outliers for each group showed that group 1 (who selected the DT system) did not produce any outliers either using IVRIE or DT system. Group 2 (who selected IVRIE) produced 24% more out-of-range space sizes in DT than in IVRIE, and group 3 (who selected both systems) had equal percentages of produced outliers in both systems. The results indicate that the impact of systems utilization on spatial decision makings based on the percentages of produced outliers in each system was compatible with participants' evaluations of the systems' effectiveness. The calculation of Spearman's correlation coefficient and subsequent significance testing for the produced

spatial outliers by the group that evaluated IVRIE as the effective system for more accurate spatial decision-making indicates a weak positive correlation coefficient ($r_s$ = 0.2). The calculated *p*-value (0.08, non-significant) indicates that there is less than a 5% chance that the strength of this relation happened by chance if the null hypothesis was true. The correlation test for the group who evaluated the effectiveness of systems equally in allowing them to create the intended spatial sense of spaces indicates a strong positive correlation coefficient ($r_s$ = 0.63) between the produced outliers in both systems and the calculated *p*-value (0.02, significant) indicates that there is less than a 5% chance that the strength of this relation happened by chance if the null hypothesis was true. Figure 16 presents the population percentages in each group and the produced outliers percentages in each system.

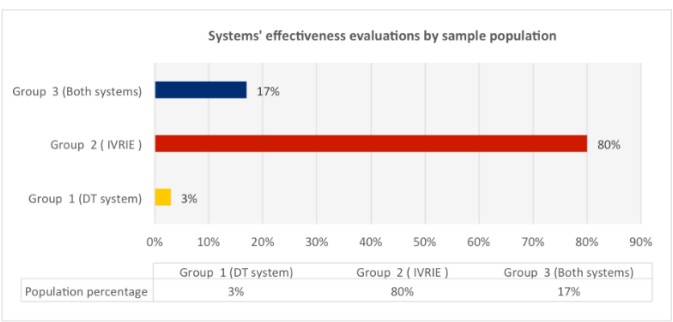 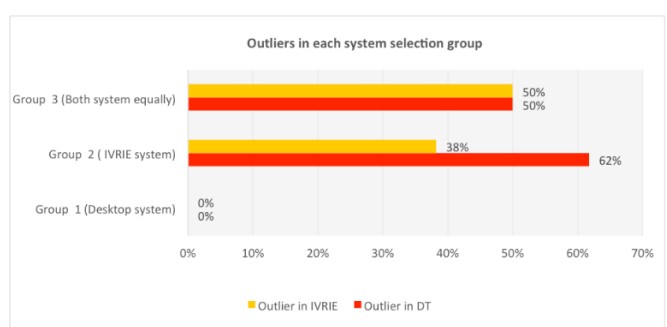

**Figure 16.** Evaluated systems' effectiveness and produced outliers in each system.

## 4. Discussion

This research explored the capability of two VR systems, IVRIE and DT systems, in enhancing user spatial awareness in a way that impacts user spatial decision-making expressed by a reduction of spatially abnormal design outcomes. Two separate branches of data were collected, analyzed, and interpreted concerning the research objectives to address the research questions. One branch of data was the quantitative extracted from the measurements of design results. The second branch consisted of qualitative data, which was coded into numeric scales for being used in statistical testing, collected through a questionnaire. The quantitative data was analyzed exclusively to identifying the percentage of design outcome outliers in each system. These data were then utilized to identify the relation between user perception of systems functionality for spatial design and systems performance in reducing the number of space size outliers.

The research tested three hypotheses. The first hypothesis proposed that specific features of IVRIE, including immersion, direct interaction, and eye-level view would facilitate user spatial perception and decrease the probability of extreme spatial decisions. The findings support this hypothesis and reveal that when using IVRIE and DT systems for spatial design, spatial decisions made by individuals using IVRIE systems generate less outliers as compared to individuals using DT system. The second hypothesis proposed that regardless of systems usage sequence, when a user gets familiar with spatial characteristics in the first system, it would result in less spatial outliers in the second system. The findings of this study do not support this hypothesis. Only the system usage sequence from IVRIE to DT positively reduced spatial outliers. The third hypothesis proposed that if users perceive one of the systems more functional for their spatial design, they would be aware of the helpfulness of that system's feature in enhancing their spatial decision-making. The findings support this hypothesis, and analyses found that the participants who ranked IVRIE as the most functional system for spatial design produced fewer spatial outliers than when using the DT systems. In addition, participants who ranked both systems functional for spatial design produced equal spatial outliers in DT system and IVRIE. The overall findings of this study can be summarized as follows:

1.  On average, the percentage of produced spatial outliers, utilizing DT system was higher than IVRIE, while there were different association levels between the space

characteristics, systems usage sequence, and the production of out-of-range space sizes in each system. The association between space types (fully enclosed space or open-ended corridor) and produced out-of–range space sizes in each system was weak. The presence/absence of texture had a medium association with the production of spatial outliers in each system. In addition, there was a weak association between the produced out-of-range space sizes in each system and the variation of systems usage sequence.

2.  Between the two tested users' characteristics, including gender and educational major, there was a weak association between genders and produced spatial outliers in each system. However, there was a medium association between participants' majors and their produced out-of-range space sizes utilizing both systems.

3.  On average, the participants who evaluated direct interaction, immersion, and having eye-level view for spatial decision-making in IVRIE "very helpful" produced fewer outliers in this system compared to the DT system.

4.  On average, the participants with systems usage sequence from IVRIE to DT produced fewer spatial outliers in both systems than those with opposite systems usage sequence. However, there is no significant correlation between the produced out-of-range space sizes in both systems by the participants who evaluated the role of systems usage sequence "very helpful" in spatial decision-making.

5.  There is a significant positive correlation between the out-of-range space sizes in both systems, produced by participants who evaluated both IVRIE and DT systems' effectiveness in allowing them to create the intended spatial sense of designed spaces.

Although the number of studies using quantitative and mixed methods for testing and identifying the IVR systems usability in architectural design learning and practice has increased, this body of research still has a long way to go to get a comprehensive perspective regarding the impacts of these systems on user spatial perception, awareness and design learning. The findings of the current study could be utilized as the first steps in identifying the impacts of IVR systems utilization on user spatial decision-making.

Although spatial/experiential guidelines were designed to be straightforward and easy to use, based on some participants' requests for more details about the spatial feelings in the spaces, the descriptions of the spatial function of spaces in the guidelines seemed to need more clarification and details. Regarding the designated spatial function for corridor spaces for three people walking, some participants asked for more information about how people are supposed to walk in the corridors, such as walking in a line or walking like a group of three friends parallel to each other. In addition, when participants were working on redesigning the models, specifically enclosed spaces, some mentioned that if the spaces had a roof, their spatial decisions would be different.

In future research, along with considering these points, other conditions and variables, such as utilizing the spatial/experiential guidelines for designing more spatially complex spaces should be explored. In addition, the spatial decisions of users when the spaces are connected and have spatial relationships or different textures are presented in one space should be tested.

**Author Contributions:** Conceptualization, S.A. and A.R.; methodology, S.A. and A.R.; validation, S.A.; formal analysis, S.A.; data curation, S.A.; visualization, S.A.; writing—original draft preparation, S.A.; writing—review and editing, S.A. and A.R. All authors have read and agreed to the published version of the manuscript.

**Funding:** This research received no external funding.

**Institutional Review Board Statement:** The study was conducted in accordance with the Declaration of Helsinki, and approved by the Institutional Review Board of NORTH CAROLINA STATE UNIVERSITY (protocol code: 21067, approval date: 18 November 2020).

**Informed Consent Statement:** Informed consent was obtained from all subjects involved in the study.

**Data Availability Statement:** The data presented in this study are available on request from the corresponding author.

**Acknowledgments:** Thanks to the students of the North Carolina State University for their generous participation in this study.

**Conflicts of Interest:** The authors declare no conflict of interest.

## Abbreviations

| | |
|---|---|
| VR | Virtual Reality |
| IVR | Immersive Virtual Reality |
| IVRIE | Immersive Virtual Reality Interactive Environment |
| DT | Desktop system |
| 3D | Three-dimensional |

## Appendix A. Spatial/Experiential Guidelines

In each scenario, there are four different spaces consisting of two spaces with two walls and two spaces with four walls. None of the spaces have roofs.
Please follow the guidelines to redesign the given spaces.

1. Design space number 1 to be wide enough for one person to comfortably walk down, but with the feeling of being confined and enclosed.

2. Design space number 2 to be a comfortable space for two people to gather that does not feel too large or too small.

3. Design space number 3 to be comfortable for three people to walk down and not feel too wide or too narrow.

4. Design space number 4 to be a space for ten people to comfortably gather that feels spacious and expansive but limited and defined.

**Appendix B. Spatial Perception Questionnaire**

**Please answer the following questions:**

1. In thinking about your understanding of the spaces you were designing, how helpful was it to directly interact with objects in VR through the handles?

   ☐ Very helpful ☐ Somewhat helpful ☐ Slightly helpful ☐ Not at all helpful

2. When making decisions about the feeling of your designed spaces, how helpful was it to you to have an eye-level view in VR?

   ☐ Very helpful ☐ Somewhat helpful ☐ Slightly helpful ☐ Not at all helpful

3. When making decisions about the feeling of your designed spaces, how helpful was it to you to be fully immersed in VR?

   ☐ Very helpful ☐ Somewhat helpful ☐ Slightly helpful ☐ Not at all helpful

4. In the experiment, you worked in the first system, got familiar with the given spaces, and then moved to the other system. How helpful was this process in enabling you to estimate the feeling of your designed spaces better when using the second system?

   ☐ Very helpful ☐ Somewhat helpful ☐ Slightly helpful ☐ Not at all helpful

5. Which system do you feel allowed you to effectively create the intended spatial sense through your designed spaces most accurately?

   ☐ Desktop system ☐ VR system ☐ Both systems equally

Thank you for your participation in this research study.

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
