# Peer review of "User Performance in Virtual Reality Environments: The Capability of Immersive Virtual Reality Systems in Enhancing User Spatial Awareness and Producing Consistent Design Results"

_sustainability, doi:10.3390/su142114129_

Round 1

Reviewer 1 Report

The paper explores how Immersive Virtual Reality Interactive Environment and a desktop-based VR system to reduce spatial decision-making based on users' spatial awareness.

The topic of the paper is original and of interest for the research community. The article is well written and well referenced, the state-of-art well presented.

I, therefore, suggest to accept the paper in the present form.

Author Response

Dear reviewer, thanks for your positive feedback and your time and consideration.

The Authors

Reviewer 2 Report

This paper contributes a study comparing whether designing spaces in immersive VR vs. a desktop-based system produces fewer size anomalies. IVR was found to have fewer anomalies. Additionally the user of surface texture was not a factor in reducing outliers and use of IVR before the DT system reduced the number of outliers.

 There are several ways the paper could be improved. First the related work section seems to be missing some important work. Almost all of the cited work seems to be from more applied venues rather than the traditional venues for foundational VR work (e.g. IEEEVR, VRST, ISMAR, etc.). Given the topic I would have expected to see some discussion of the perceptual distance compression issue in IVR (see e.g. Interrante, Victoria, Brian Ries, and Lee Anderson. "Distance perception in immersive virtual environments, revisited." In IEEE virtual reality conference (VR 2006), pp. 3-10. IEEE, 2006.). This seems like it could be a confounding factor in the experiment. The following reference also seems pertinent:

 Usman, Muhammad, Brandon Haworth, Glen Berseth, Mubbasir Kapadia, and Petros Faloutsos. "Understanding spatial perception and visual modes in the review of architectural designs." In Proceedings of the ACM SIGGRAPH/Eurographics Symposium on Computer Animation, pp. 1-2. 2017.

 Second the methodology section is missing some details that are typically reported. For example, it is common to report statistics on participant age and gender distributions. It would also be easier to parse this section if it was reorganized into subsections for design, participants, and measures.

 In the results section it is not clear how the IQR was used to detect outliers. Did the authors use 1.5xIQR or some other technique to decide the outlier fence?

 I’m less familiar with comparing outliers than traditional mean/stdev statistics, but in several placed the authors describe how the percentage differences were significant. Were any actual statistical tests run to test this? If so they should be reported. In addition to the percentages reported throughout the paper, it would also be useful to see the raw counts.

 In the discussion section, it lists that surface texture is not a significant factor in reducing spatial outliers. It seem like that claim is hard to generalized since it may just depend on the texture that was chosen. I would suggest weakening the claim. There is also some discussion about the effects of age on performance but no results for these appear earlier in the results section!

 The references have several issues. Some are missing the publication journal or page numbers. The formatting is not consistent

Author Response

Point 1: This paper contributes a study comparing whether designing spaces in immersive VR vs. a desktop-based system produces fewer size anomalies. IVR was found to have fewer anomalies. Additionally the user of surface texture was not a factor in reducing outliers and use of IVR before the DT system reduced the number of outliers.

There are several ways the paper could be improved. First the related work section seems to be missing some important work. Almost all of the cited work seems to be from more applied venues rather than the traditional venues for foundational VR work (e.g. IEEEVR, VRST, ISMAR, etc.). Given the topic I would have expected to see some discussion of the perceptual distance compression issue in IVR (see e.g. Interrante, Victoria, Brian Ries, and Lee Anderson. "Distance perception in immersive virtual environments, revisited." In IEEE virtual reality conference (VR 2006), pp. 3-10. IEEE, 2006.). This seems like it could be a confounding factor in the experiment. The following reference also seems pertinent:

 Usman, Muhammad, Brandon Haworth, Glen Berseth, Mubbasir Kapadia, and Petros Faloutsos. "Understanding spatial perception and visual modes in the review of architectural designs." In Proceedings of the ACM SIGGRAPH/Eurographics Symposium on Computer Animation, pp. 1-2. 2017.

Response 1: Dear reviewer, thanks for all your recommendations and comments. Based on your suggestions, the mentioned related works were reviewed and discussed in the article’s ‘research background’ section.

Point 2:Second the methodology section is missing some details that are typically reported. For example, it is common to report statistics on participant age and gender distributions. It would also be easier to parse this section if it was reorganized into subsections for design, participants, and measures.

Response 2 : This article and its related research are not focused on participants’ characteristics, and research objectives were designed to explore the capability of these two virtual reality systems (DT system and IVRIE) in reducing the extreme variations in user spatial decision-making. The authors explored the impact of user characteristics on spatial perception in IVR systems in a recently published article. In that study, the impacts of user characteristics such as age, gender, educational level, professional design experience, and levels of familiarity with 3D environments in both DT and IVRIE were explored. Please see “Azarby, Sahand, and Arthur Rice if interested. “Scale Estimation for Design Decisions in Virtual Environments: Understanding the Impact of User Characteristics on Spatial Perception in Immersive Virtual Reality Systems.” Buildings 12.9 (2022): 1461.”

 In the results section it is not clear how the IQR was used to detect outliers. Did the authors use 1.5xIQR or some other technique to decide the outlier fence?

I’m less familiar with comparing outliers than traditional mean/stdev statistics, but in several placed the authors describe how the percentage differences were significant. Were any actual statistical tests run to test this? If so they should be reported. In addition to the percentages reported throughout the paper, it would also be useful to see the raw counts.

Response 3 : You are right; when we calculate the IQR, which is the subtraction of the first and third quartiles (Q1 and Q3), any data point which lies below the Q1 – 1.5 IQR or above Q1 + 1.5 IQR is an outlier. This calculation was used to detect all the out-of-range space sizes designed by the sample population in both systems. A sentence for more clarification is added to the first paragraph of the ‘results’ section.

In the 'results' section, in just one case, the percentages of produced out-of-range fully enclosed spaces between the two systems were compared (50% versus 18.2%); the result of comparison and level of difference was called 'significantly different'. In statistical language, when comparing two samples with equal number of members based on a common variable and the difference is around 30% or above, it is fine to describe it as a significant difference. For the statistical comparisons in this research, we did not use any specific statistical tests such as two-sample t-test or two-sample F-test since the first step for normalization of the samples' distributions for these tests is the detection and removal of outliers (bad data) which was the main variable for all the analyses of this study.

 In the discussion section, it lists that surface texture is not a significant factor in reducing spatial outliers. It seem like that claim is hard to generalized since it may just depend on the texture that was chosen. I would suggest weakening the claim. There is also some discussion about the effects of age on performance but no results for these appear earlier in the results section!

Response 4 :The claim about the role of texture is weakened, and the word ‘impactful’ is replaced. Any discussion regarding the age or users’ specifications is removed.

 The references have several issues. Some are missing the publication journal or page numbers. The formatting is not consistent

Response 5 :The references are reviewed and updated.

Many thanks for your time and consideration.

The Authors

Reviewer 3 Report

The article presents a very interesting topic especially in the field of architectural design, given the proliferation of immersive virtual environments and recent studies on the Metaverse. The research methodology is well explained as well as the implementation of the immersive technology so that it can be easily replicated. The experimental results are very clear and devoid of ambiguity. 

The article is about the level of spatial awareness one has within a virtual environment. Specifically, the experiments described compare spatial decision-making ability in an immersive virtual environment and a desktop-based one. The article is well written and has a structure that does not lead to confusion. The approach is certainly novel in that low-cost tools and software are used. The conclusions show in great detail the superiority of the immersive system in avoiding errors. 

Author Response

(The authors gave the same response as above.)

Round 2

Reviewer 2 Report

Authors’ response: “This article and its related research are not focused on participants’ characteristics, and research objectives were designed to explore the capability of these two virtual reality systems (DT system and IVRIE) in reducing the extreme variations in user spatial decision-making. The authors explored the impact of user characteristics on spatial perception in IVR systems in a recently published article. In that study, the impacts of user characteristics such as age, gender, educational level, professional design experience, and levels of familiarity with 3D environments in both DT and IVRIE were explored. Please see “Azarby, Sahand, and Arthur Rice if interested. “Scale Estimation for Design Decisions in Virtual Environments: Understanding the Impact of User Characteristics on Spatial Perception in Immersive Virtual Reality Systems.” Buildings 12.9 (2022): 1461.”

Regardless of whether the particular research questions studied involved participant characteristics, they have an effect on the results (As shown in the cited prior work that the authors mention!). It is standard that they be reported and statistical analysis run to show that the results are in fact only due to the differing technology and not because the different participant groups contained more subjects of a particular age, gender, experience level, etc.

Authors’ Response: “For the statistical comparisons in this research, we did not use any specific statistical tests such as two-sample t-test or two-sample F-test since the first step for normalization of the samples' distributions for these tests is the detection and removal of outliers (bad data) which was the main variable for all the analyses of this study.

This is actually one of the largest issues I have with this paper. Because no statistics were run, it is very hard to justify whether the results are just due to chance or would be replicated in further studies. At the very least, something like a Cramer’s V correlation test should be run by categorizing the dependent variable as either an outlier or not. Similarly for the likert analysis, Spearman correlation tests could be run. It is hard to justify publication without some actual analysis that the results are in fact correct.

Finally, some of the references formatting is still odd (e.g. all caps titles). In general, you shouldn’t blindly trust the bibtex downloads from digital library sites without editing them for formatting and missing information.

Author Response

Response to Reviewer 2 Comments

Point 1: Regardless of whether the particular research questions studied involved participant characteristics, they have an effect on the results (As shown in the cited prior work that the authors mention!). It is standard that they be reported and statistical analysis run to show that the results are in fact only due to the differing technology and not because the different participant groups contained more subjects of a particular age, gender, experience level, etc.

Response 1: Dear reviewer, thanks for your recommendations and comments. Based on your suggestions, participants' gender and educational majors as user characteristics are included in the analyses, and their associations with the production of spatial outliers in both IVRIE and DT systems were tested. The users' age could not be included in the analyses since all participants' ages were between 20 to 25 years, and no additional data was collected for its distinction.

Point 2:This is actually one of the largest issues I have with this paper. Because no statistics were run, it is very hard to justify whether the results are just due to chance or would be replicated in further studies. At the very least, something like a Cramer’s V correlation test should be run by categorizing the dependent variable as either an outlier or not. Similarly for the likert analysis, Spearman correlation tests could be run. It is hard to justify publication without some actual analysis that the results are in fact correct.

Response 2: Based on your recommendations, in the first part of the results section (3.1. Percentage of out of range design results within each system), all the findings were tested and analyzed through the application of Cramer’s V test and the strength of associations between the produced spatial outliers in both systems with other variables, including spaces characteristics, users’ characteristics and systems usage sequence identified. In addition, for all of the Likert analyses in the second part of the results section (3.2. Percentage of space size outliers and systems efficiency evaluation ), the correlations were tested and analyzed using Spearman’s correlation test.

Point 3:Finally, some of the references formatting is still odd (e.g. all caps titles). In general, you shouldn’t blindly trust the bibtex downloads from digital library sites without editing them for formatting and missing information.

Response 3 :The references are reviewed and updated.

Many thanks for your time and consideration.

The Authors

Round 3

Reviewer 2 Report

The authors have adequately addressed my concerns.